# Blood pressure variability and night-time dipping assessed by 24-hour ambulatory monitoring: Cross-sectional association with cardiac structure in adolescents

Lucy J. Goudswaard[1,2,3]*, Sean Harrison[2,3], Daniel Van De Klee[4], Nishi Chaturvedi[5], Debbie A. Lawlor[2,3], George Davey Smith[2,3], Alun D. Hughes[5], Laura D. Howe[2,3]

1 School of Physiology, Pharmacology and Neuroscience at the University of Bristol, Bristol, United Kingdom, 2 MRC Integrative Epidemiology Unit at the University of Bristol, Bristol, United Kingdom, 3 Population Health Sciences, Bristol Medical School, University of Bristol, Bristol, United Kingdom, 4 Acute GP Team, BrisDoc Healthcare Services, Bristol, United Kingdom, 5 Institute of Cardiovascular Science, University College London, London, United Kingdom

* lg14289@bristol.ac.uk

**Data Availability Statement:** Data used for analyses contain potentially identifying and

## Abstract

Greater blood pressure (BP) is associated with greater left ventricular mass indexed to height$^{2.7}$ (LVMi$^{2.7}$) in adolescents. This study examined whether greater BP variability and reduced night-time dipping are associated with cardiac remodeling in a general population of adolescents. A cross-sectional analysis was undertaken in 587 UK adolescents (mean age 17.7 years; 43.1% male). BP was measured in a research clinic and using 24-hour ambulatory monitoring. We examined associations (for both systolic and diastolic BP) of: 1) clinic and 24-hour mean BP; 2) measures of 24-hour BP variability: standard deviation weighted for day/night (SDdn), variability independent of the mean (VIM) and average real variability (ARV); and 3) night-time dipping with cardiac structures. Cardiac structures were assessed by echocardiography: 1) LVMi$^{2.7}$; 2) relative wall thickness (RWT); 3) left atrial diameter indexed to height (LADi) and 4) left ventricular internal diameter in diastole (LVIDD). Higher systolic BP was associated with greater LVMi$^{2.7}$. Systolic and diastolic BP were associated with greater RWT. Associations were inconsistent for LADi and LVIDD. There was evidence for associations between both greater SDdn and ARV and higher RWT (per 1 SD higher diastolic ARV, mean difference in RWT was 0.13 SDs, 95% CI 0.045 to 0.21); these associations with RWT remained after adjustment for mean BP. There was no consistent evidence of associations between night-time dipping and cardiac structure. Measurement of BP variability, even in adolescents with blood pressure in the physiologic range, might benefit risk of cardiovascular remodeling assessment.

sensitive information, therefore access needs to be approved by the ALSPAC Executive Committee, who can be contacted at alspac-data@bristol.ac.uk.

**Funding:** The UK Medical Research Council and Wellcome (Grant ref: 102215/2/13/2) and the University of Bristol provide core support for ALSPAC. LJG is funded by a University of Bristol alumni PhD studentship as part of the British Heart Foundation 4-year Integrative Cardiovascular Science programme. LDH is funded by a Career Development Award from the UK Medical Research Council (MR/M020894/1). LDH, SH, DAL and GDS work in a unit that receives funding from the University of Bristol and the UK Medical Research Council (MC_UU_00011/1 and MC_UU_00011/6-7). This work was supported by a grant from the British Heart Foundation. The funders had no role in study design, data collection and analysis, decision to publish, or preparation of the manuscript. For the purpose of Open Access, the author has applied a CC BY public copyright licence to any Author Accepted Manuscript version arising from this submission.

**Competing interests:** The authors have declared that no competing interests exist.

## Introduction

Higher blood pressure (BP) is associated with an increased risk of cardiovascular disease (CVD) [1]. However, BP is inherently variable, and under a typical circadian rhythm night-time BP is lower than daytime [2]. Loss of this nocturnal dipping pattern in the general population of adults has been shown to be associated with cardiovascular events and all-cause mortality, independent of 24-hour BP [2,3]. There is also evidence that non-circadian variability in BP may be associated with cardiovascular disease [2,4,5].

Cardiovascular pathology starts in early life, with childhood BP levels known to track across life [6], and early adulthood BP relating to mortality from CVD [7]. In adults, higher left ventricular (LV) mass and left atrial enlargement are both associated with higher risk of CVD [8,9] and are considered evidence of target organ damage [10]. Another measure of left heart function, relative wall thickness (RWT, a measure of remodeling [11]), has been suggested to be predictive of stroke among adult populations [12,13]. We previously demonstrated that in 17 year-olds that higher body mass index (BMI) is causally related to higher LV mass indexed to height$^{2.7}$ (LVMi$^{2.7}$) [14], suggesting that there is meaningful variation in cardiac structure measures in early adulthood. A study in adults from the general population indicated a positive association between BP variability and LVMi [15]. Associations between BP variability and cardiac structures in children with suspected hypertension have been explored [16], but it is unclear if any associations are apparent in a general population of adolescents.

In this study, we used data from a prospective cohort study of 587 UK adolescents to assess the cross-sectional associations of mean BP (from clinic measurements and ambulatory monitoring), BP variability, and night-time dipping, with measures of cardiac structure at age 17, determined by echocardiography. The measures of cardiac structure we consider are 1) LV mass (LVM), 2) RWT [11], 3) left atrial diameter (LAD), and 4) left ventricular internal diameter during diastole (LVIDD, a measure of the initial stretching of cardiomyocytes before contraction (preload)) [17]. Together these represent a comprehensive assessment of left heart structure, with functional significance [18].

## Methods

### Participants

ALSPAC is a population-based birth cohort. The study recruited pregnant women from the Avon area (Bristol) in the South West of England, with an expected delivery date between 1$^{st}$ April 1991 and 31$^{st}$ December 1992 [19]. From the 15,643 pregnant women enrolled, 14,889 children were born and alive at one year [19,20] (Fig 1). Since birth, participants have been followed up, using questionnaires, links to routine data, and research clinics. The study website provides further details of the cohort and a data dictionary http://www.bris.ac.uk/alspac/researchers/data-access/data-dictionary/. Ethical approval was obtained from the local ethics committee and the ALSPAC Law and Ethics committee. Written informed consent was obtained from parents and children were invited to give verbal assent where appropriate. Participants were able to withdraw at any time.

### Inclusion/Exclusion criteria

This was a cross-sectional study conducted in participants who attended the 17-year follow-up clinic of ALSPAC. Participants were eligible if they attended both the echocardiography and the 24-hour blood pressure sub-studies at the 17-year clinic visit. We a priori decided to exclude participants if they were pregnant or reported taking antihypertensive medication or having a congenital cardiac anomaly, but this did not apply to any participants in the study.

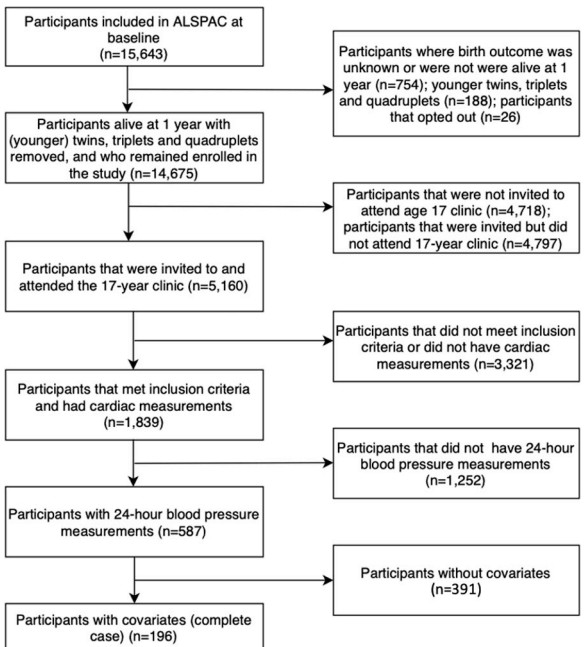

**Fig 1. A STROBE diagram detailing how the study cohort was selected from the baseline Avon Longitudinal Study of Parents And Children (ALSPAC) participants.**

## Exposures

**1) Clinic and 24-hour ambulatory blood pressure measurements.** Clinic systolic blood pressure (SBP) and diastolic blood pressure (DBP) were measured with an OMRON 705 IT oscillometric BP monitor (Omron Corporation, Kyoto, Japan) with the participant sitting and at rest with their arm supported. Readings were taken in accordance with European Society of Hypertension guidelines [21]. We used the average of the final two measures from the right arm in our analyses.

Participants were fitted with a 24-hour ambulatory blood pressure monitor (ABPM) (Space-labs 90217, Washington, U.S.) according to the manufacturer's instructions. This measured their brachial BP, with readings taken every 30 minutes during the day and hourly at night. Participants were permitted to perform usual physical activities, although a diary of activities was recorded. Daytime and night-time were defined by the participant. The expected maximum number of total readings per participant therefore varied depending on the duration of the night-time period. For this study, we included participants with at least 14 readings during the self-defined daytime and at least 5 readings during the self-defined night-time [22,23]. We estimated the mean 24-hour SBP and DBP using the ABPM data, and also estimated the daytime and night-time means for SBP and DBP.

**2) Measures of blood pressure variability.** We estimated variability in the 24-hour systolic and diastolic measures in three different ways. 1) Standard deviation weighted for daytime and night-time (SDdn) [24], calculated as: $\frac{(\text{day SD} \times \text{day hours}) + (\text{night SD} \times \text{night hours})}{\text{day hours} + \text{night hours}}$. 2) Average real variability (ARV), derived as the average of the differences between consecutive BP measurements [25], using the formula: $\frac{1}{N-1}\sum_{k=1}^{N-1}|BP_{k+1} - BP_k|$, where N is the number of valid blood pressure (BP) readings and k is the number of the individual reading. To derive

this variable, each individual blood pressure reading and the order of readings was required (at least 14 daytime readings and 5 night-time readings). 3) Variability independent of the mean (VIM) [26], derived using the formula: $\frac{SDdn}{mean^x} \times population\ mean^x$, where $x$ is derived from the regression coefficient β from the equation: $\ln(SD) = \alpha + \beta \ln(mean)$.

**3) Dipping variables.** We estimated night-time dipping as a percentage difference between daytime and night-time means ($\frac{24hr\ daytime\ BP - 24hr\ nighttime\ BP}{24hr\ daytime\ BP} \times 100$) [27]. We considered participants with ≥10% reduction in night-time BP compared to daytime BP as 'normal dippers', and those with <10% reduction or an increase as 'non-dippers' in a binary dipping variable [3,27]. We also grouped participants into four dipping groups of 1) Normal dippers (>10%, ≤20%), 2) Non-dippers (>0%, ≤10%), 3) Extreme dippers (>20%) and 4) Risers (<0%) to allow comparison of dipping distribution with previous studies [28], however only the simpler continuous and dichotomised variables are included as an exposure in our analyses because of the relatively small sample size.

## Outcomes: Echocardiography measurements

Echocardiography was performed on a quasi-random subsample (based on date of research clinic attendance) using an HDI 5000 ultrasound machine (Philips, Massachusetts, U.S.) equipped with a P4-2 Phased Array ultrasound transducer. One of two echocardiographers examined participants using a standard examination protocol, in accordance with the American Society of Echocardiography (ASE) guidelines [29]. All measures were made in end diastole and were calculated as the mean of three measurements. LV mass was calculated from end-diastolic ventricular septal wall thickness (SWTd), left ventricular dimension (LVIDd), and left ventricular posterior wall thickness (PWT) according to the ASE formula: $0.8 \times (1.04 \times [(SWT + LVIDD + PWT)^3 - (LVIDD)^3]) + 0.6$. LV mass was then indexed to height$^{2.7}$ (LVMi$^{2.7}$) using the Troy formula in order to account for differences in body sizes [30]. Left atrial diameter was indexed to height [9] (LADi). Relative wall thickness (RWT) was calculated using the formula: $\frac{PWT + SWT}{LVIDD}$.

## Confounders

We considered variables as confounders if they had plausible relations with BP and cardiovascular risk [31]. Maternal confounders were self-reported in questionnaires completed during pregnancy: educational attainment (categorised as university degree or higher, Advanced-levels (exams usually taken around 18 years and necessary for university entry), Ordinary-levels (exams usually taken around 16 years, which was the minimum UK school leaving age at the time these participants were this age), or lower than Ordinary-levels, including vocational education); pre-pregnancy body mass index (BMI; in kg/m$^2$); age at delivery (categorised as <25 years, 25–35 years, and >35 years); parity, and highest head of household occupational social class. We selected these maternal variables as the mother's socioeconomic position (SEP) represents the participant's family SEP. SEP has been shown to influence BMI (a key determinant of both BP and LVM [14]), blood pressure [32], and left ventricular structure [33]. Maternal pre-pregnancy BMI has also been shown to affect offspring BP and cardiovascular outcomes [34].

Child-based confounders were from a combination of self-reported questionnaire and clinic-based data. These include: age (in months) at year 17 clinic visit; smoking at age 17 (<1 or ≥1 cigarette per week from self-report); minutes of moderate to vigorous physical activity at age 15 assessed by uniaxial ActiGraph accelerometer (Florida, U.S.) and used as quintiles in the analysis; percentage fat mass (assessed by dual energy-X-ray absorptiometry (DXA) at the

17-year clinic using a Lunar prodigy narrow fan beam densitometer); and height measured at the 17-year clinic using a Harpenden stadiometer (Holtain Ltd, Crymych, UK). These child-based variables were selected as they likely affect cardiovascular health [14].

## Statistical analysis

All analyses were performed using Stata version 15.1 (StataCorp, TX).

We used multivariable linear regression to estimate the associations between all blood pressure exposures and cardiac structure outcomes defined above. We standardised all exposures and outcomes before analysis to have a mean of zero and SD of one. As such, all regression results are interpreted as the SD change in the outcome for a SD change in the exposure. For the binary dipping variables, the regression result can be interpreted as the change in outcome variable in SDs comparing the non-dippers category with the dippers.

Associations between each of the 18 BP exposures (for both SBP and DBP: clinic BP, 24h mean BP, mean daytime BP, mean night-time BP, SDdn, ARV, VIM and continuous and binary dipping variables) and 4 measures of cardiac structure (LVMi$^{2.7}$, LADi, RWT, LVIDD) were assessed using multivariable linear regression. Three models were estimated: i) adjustment for sex and age at year 17 clinic visit, ii) additional adjustment for potential confounders: maternal education, age at delivery, parity, pre-pregnancy BMI; household socio-economic class; smoking at age 17; minutes of moderate to vigorous physical activity at age 15; DXA-determined fat mass and height and height$^2$ at age 17, iii) further adjustment for average 24-hour blood pressure (systolic or diastolic as appropriate for the exposure) to evaluate whether any associations between BP variability and dipping were independent of 24-hour average BP. To verify analyses were not affected by collinearity, we assessed correlations between measures of mean BP and measures of BP variability.

To test for interactions between sex and each exposure, we regressed each outcome on each exposure, with sex and an interaction term for the exposure and sex as covariables. There was no strong evidence of any interactions by sex from these analyses (p>0.1 for all interaction terms), and as such, all results are presented for males and females combined. To check for linearity of a) blood pressure—cardiac structure and b) mean blood pressure–blood pressure variability associations, we conducted likelihood ratio tests comparing models where fifths of the exposure variable were treated as numeric and categorical variables. There was no evidence of non-linearity in the associations between blood pressure variables and cardiac structure outcomes, and so results are presented with continuous measures of blood pressure as the exposures. As a sensitivity analysis and to account for nonlinearity in the associations between blood pressure and blood pressure variability, the association between blood pressure variability/dipping exposures and cardiac structure outcomes were also explored adjusting for categorical fifths of average blood pressure.

We did not correct the results for multiple testing, as multiple testing correction emphasises the inappropriate dichotomisation of p-values into significant versus non-significant [35–38]. Furthermore, in this analysis, exposures are correlated measures of a single underlying construct BP, and outcomes are measures of a single underlying construct, cardiac structure. A Bonferroni multiple testing correction would therefore be over-conservative. We interpret the overall pattern of results rather than focusing on single p-values, and use the magnitude of coefficients and confidence intervals to assess the strength of associations.

## Missing data

Of the 587 participants with complete data on all 18 exposures and 4 outcomes, 196 (33.3%) also had complete data including all confounders. In the full dataset, individual confounder

variables were missing between 0% and 43.4% of observations, with eight of 11 variables having less than 13% missingness (S1a Table in S1 File). We used multivariate multiple imputation by chained equations to impute missing confounder data [39,40]. The imputation model included all exposures (excluding dipping variables, which were derived from other variables in the imputation model), outcomes and confounding variables, as well as weight and BMI at age 17, and maternal height. Fully conditional specification was used, with linear regression for continuous variables, multinomial regression for categorical variables and logistic regression for binary variables (S1b Table in S1 File). We created twenty imputed datasets and used Rubin's rules to combine analysis results. Variable distributions were consistent between the imputed and the observed data sets (S1a Table in S1 File). We also conducted a complete case sensitivity analysis in the 196 participants with complete data for all variables (S2 Table in S1 File).

## Results

### Participant characteristics

A total of 587 participants of European ancestry were included in our analysis. Fig 1 shows how this cohort size was reached from the participants enrolled in ALSPAC at baseline. Compared with the full ALSPAC cohort, the participants included in our analysis tended to have mothers who were more educated and older when the participant was born and be from a family with a higher head of household occupational social class; females were also more likely to be included. Clinic blood pressure, minutes of moderate to vigorous physical activity at age 15 and DXA-determined fat mass were similar compared with the full ALSPAC cohort (S1b Table in S1 File).

Of the included participants, 43.1% were male, mean age was 17.7 (SD 0.3) years, 2.1% reported smoking 1 or more cigarettes a week. The mean BMI was 23.1 (SD 4) kg/m$^2$. Mean clinic systolic and diastolic blood pressure were 114.4 mmHg (SD 9.7 mmHg) and 64.5 mmHg (SD 5.8 mmHg), respectively (Table 1).

Males tended to have higher systolic blood pressure, pulse pressure and mean arterial pressure, while females had higher diastolic blood pressure. Night-time dipping was similar between sexes. Males tended to have higher systolic and diastolic BP variability than females. Ventricular measures were higher in males, while LADi and RWT were similar between sexes (Table 1).

### Associations between clinic BP measurements and cardiac structures

Clinic SBP was associated with higher LVMi$^{2.7}$ (β = 0.23 SDs per SD increase in SBP, 95% CI 0.15 to 0.32, P = 1.6x10$^{-7}$) and higher RWT (β = 0.29 SDs per SD increase in SBP, 95% CI 0.19 to 0.39, P = 1.2x10$^{-8}$) after adjustment for confounders (Table 2). There was no evidence of associations with LADi or LVIDD.

Clinic DBP was associated with higher RWT (β = 0.24, 95% CI 0.15 to 0.33, P = 1.4x10$^{-7}$) and lower LADi and LVIDD. There was no evidence of an association between clinic DBP and LVMi$^{2.7}$.

Results were broadly similar in the age and sex only adjusted models (S4 Table in S1 File).

### Associations between ambulatory averages of BP and cardiac structures

There was evidence for a positive association between 24-hour mean SBP and LVMi$^{2.7}$ (β = 0.17 SDs per SD higher 24-hour SBP, 95% CI 0.093 to 0.25, P = 1.8x10$^{-5}$, Table 2), which was slightly smaller in magnitude than the association for clinic SBP (Fig 2A). Daytime and night-time means for SBP also showed positive associations with LVMi$^{2.7}$, with similar magnitudes

**Table 1. Measures of blood pressure and cardiac structure for participants included in the analysis, N = 587.**

| Variable | Combined mean (SD) or frequency (%) N = 587 | Mean (SD) or frequency (%) in males N = 253 | Mean (SD) or frequency (%) in females N = 334 | P value for sex difference |
|---|---|---|---|---|
| *Systolic Blood Pressure* | | | | |
| Clinic SBP (mmHg) | 114.5 (9.7) | 119.7 (8.9) | 110.5 (8.3) | <0.001 |
| 24h average SBP (mmHg) | 118.3 (8.6) | 121.4 (7.7) | 115.9 (8.4) | <0.001 |
| Daytime average SBP (mmHg) | 124.8 (9.2) | 128.3 (8.5) | 122.1 (8.8) | <0.001 |
| Night time average SBP (mmHg) | 107.4 (9.2) | 109.8 (8.9) | 105.6 (9.0) | 0.001 |
| SDdn SBP (mmHg) | 10.2 (2.1) | 10.6 (2.1) | 9.9 (2.1) | <0.001 |
| VIM of SBP (mmHg) | 10.2 (2.0) | 10.3 (1.9) | 10.1 (1.7) | P = 0.14 |
| ARV of SBP (mmHg) | 10.5 (2.5) | 10.9 (2.5) | 10.1 (2.5) | <0.001 |
| Systolic dipping (%) | 13.8 (5.7) | 14.4 (5.8) | 13.4 (5.5) | 0.05 |
| Binary systolic dipping: | | | | 0.20* |
| •Dippers (>10%) | 456 (77.7%) | 203 (80.2%) | 253 (75.8%) | |
| •Non-dippers (≤10%) | 131 (22.3%) | 50 (19.8%) | 81 (24.3%) | |
| Categorical systolic dipping: | | | | 0.11 |
| •Normal dippers (>10%, ≤20%) | 216 (64.7%) | 160 (63.2%) | 216 (64.7%) | |
| •Non-dippers (0–10%) | 78 (23.4%) | 46 (18.2%) | 78 (23.4%) | |
| •Extreme dippers (>20%) | 37 (11.1%) | 43 (17.1%) | 37 (11.1%) | |
| •Risers (<0%) | 3 (0.9%) | 4 (1.7%) | 3 (0.9%) | |
| *Diastolic Blood Pressure* | | | | |
| Clinic DBP (mmHg) | 64.5 (5.8) | 63.3 (5.3) | 65.4 (6.1) | <0.001 |
| 24h mean DBP (mmHg) | 67.9 (5.2) | 67.5 (5.1) | 68.1 (5.3) | 0.15 |
| Daytime average DBP (mmHg) | 73.5 (5.9) | 73.1 (5.9) | 73.9 (5.9) | 0.13 |
| Night time average DBP (mmHg) | 58.3 (5.5) | 57.9 (5.3) | 58.7 (5.6) | 0.07 |
| SDdn DBP (mmHg) | 8.4 (1.8) | 8.7 (1.9) | 8.2 (1.7) | 0.001 |
| VIM of DBP (mmHg) | 8.4 (1.8) | 8.8 (1.9) | 8.1 (1.7) | <0.001 |
| ARV of DBP (mmHg) | 8.8 (2.0) | 9.1 (2.1) | 8.6 (1.9) | 0.005 |
| Diastolic dipping (%) | 20.5 (6.9) | 20.7 (6.8) | 20.4 (7.0) | 0.59 |
| Binary diastolic dipping: | | | | 0.99* |
| •Dippers (>10%) | 550 (93.7%) | 237 (93.7%) | 313 (93.7%) | |
| •Non-dippers (≤10%) | 37 (6.3%) | 16 (6.3%) | 21 (6.3%) | |
| Categorical diastolic dipping: | | | | 0.91 |
| •Normal dippers (>10%, ≤20%) | 218 (37.1%) | 91 (36.0%) | 127 (38.2%) | |
| •Non-dippers (0–10%) | 31 (5.3%) | 14 (5.5%) | 17 (5.1%) | |
| •Extreme dippers (>20%) | 332 (56.6%) | 146 (57.7%) | 186 (55.7%) | |
| •Risers (<0%) | 6 (1.0%) | 2 (0.8%) | 4 (1.2%) | |
| *Cardiac structure measures* | | | | |
| LVMi (g/m$^{2.7}$) | 27.7 (5.9) | 29.3 (6.2) | 26.5 (5.4) | <0.001 |
| LADi (cm/m) | 1.88 (0.22) | 1.87 (0.23) | 1.88 (0.22) | 0.42 |
| LVIDD (cm) | 4.52 (0.44) | 4.76 (0.41) | 4.33 (0.36) | <0.001 |
| RWT | 0.37 (0.06) | 0.37 (0.05) | 0.37 (0.06) | 0.92 |

* = using Pearson's Chi-Squared test for the categorical dipping variable; SBP = systolic blood pressure, SDdn = standard deviation weighted for day and night, ARV = average real variability, VIM = variability independent of the mean, DBP = diastolic blood pressure, LVMi$^{2.7}$ = left ventricular mass indexed to height$^{2.7}$, LADi = left atrial diameter indexed to height, LVIDD = left ventricular internal diameter during diastole, RWT = relative wall thickness.

**Table 2. Associations of blood pressure measurements with cardiac structure, N = 587.**

| Exposure | Mean difference in cardiac structure measures (SDs) per SD higher BP: β, 95% confidence interval, P value | | | |
|---|---|---|---|---|
| | LVMi | LADi | LVIDD | RWT |
| SBP | 0.23 (0.15 to 0.32) | 0.055 (-0.039 to 0.15) | -0.0043 (-0.085 to 0.077) | 0.29 (0.19 to 0.39) |
| | P = $1.6 \times 10^{-7}$ | P = 0.25 | P = 0.92 | P = $1.2 \times 10^{-8}$ |
| 24h mean SBP | 0.17 (0.093 to 0.25) | -0.006 (-0.088 to 0.076) | 0.016 (-0.056 to 0.087) | 0.18 (0.089 to 0.26) |
| | P = $1.8 \times 10^{-5}$ | P = 0.89 | P = 0.67 | P = $8.1 \times 10^{-5}$ |
| Daytime mean SBP | 0.17 (0.097 to 0.25) | 0.018 (-0.065 to 0.10) | 0.026 (-0.046 to 0.098) | 0.16 (0.073 to 0.25) |
| | P = $1.2 \times 10^{-5}$ | P = 0.68 | P = 0.48 | P = $3.5 \times 10^{-4}$ |
| Night-time mean SBP | 0.12 (0.042 to 0.19) | -0.016 (-0.096 to 0.064) | -0.025 (-0.095 to 0.044) | 0.18 (0.10 to 0.27) |
| | P = $2.3 \times 10^{-3}$ | P = 0.70 | P = 0.48 | P = $2.2 \times 10^{-5}$ |
| SDdn SBP | 0.073 (-0.003 to 0.15) | P = 0.019 (-0.060 to 0.098) | -0.019 (-0.088 to 0.050) | 0.15 (0.061 to 0.23) |
| | P = 0.060 | P = 0.63 | P = 0.59 | P = $7.9 \times 10^{-4}$ |
| VIM of SBP | 0.018 (-0.058 to 0.093) | 0.031 (-0.047 to 0.11) | -0.022 (-0.091 to 0.047) | 0.087 (0.002 to 0.17) |
| | P = 0.65 | P = 0.44 | P = 0.53 | P = 0.046 |
| ARV of SBP | 0.091 (0.016 to 0.17) | 0.067 (-0.012 to 0.14) | 0.010 (-0.058 to 0.078) | 0.12 (0.039 to 0.21) |
| | P = 0.017 | P = 0.096 | P = 0.77 | P = $4.3 \times 10^{-3}$ |
| Systolic dipping (continuous) | 0.033 (-0.043 to 0.11) | 0.038 (-0.042 to 0.12) | 0.060 (-0.010 to 0.13) | -0.069 (-0.15 to 0.017) |
| | P = 0.39 | P = 0.35 | P = 0.09 | P = 0.12 |
| Binary systolic dipping (non-dippers versus dippers) | -0.10 (-0.28 to 0.080) | -0.073 (-0.26 to 0.12) | -0.039 (-0.20 to 0.13) | -0.0005 (-0.20 to 0.20) |
| | P = 0.28 | P = 0.45 | P = 0.64 | P = 0.99 |
| DBP | 0.034 (-0.046 to 0.11) | -0.10 (-0.19 to -0.015) | -0.13 (-0.20 to -0.055) | 0.24 (0.15 to 0.33) |
| | P = 0.41 | P = 0.021 | P = $5.7 \times 10^{4}$ | P = $1.4 \times 10^{-7}$ |
| 24h mean DBP | 0.050 (-0.024 to 0.12) | -0.075 (-0.15 to 0.0006) | -0.055 (-0.12 to 0.013) | 0.13 (0.045 to 0.21) |
| | P = 0.18 | P = 0.052 | P = 0.11 | P = $2.6 \times 10^{-3}$ |
| Daytime mean DBP | 0.059 (-0.015 to 0.1) | -0.046 (-0.12 to 0.032) | -0.053 (-0.12 to 0.015) | 0.14 (0.052 to 0.22) |
| | P = 0.12 | P = 0.24 | P = 0.12 | P = $1.5 \times 10^{-3}$ |
| Night-time mean DBP | 0.021 (-0.053 to 0.094) | -0.081 (-0.16 to -0.003) | -0.074 (-0.14 to -0.007) | 0.13 (0.044 to 0.21) |
| | P = 0.58 | P = 0.042 | P = 0.03 | P = $2.8 \times 10^{-3}$ |
| SDdn DBP | 0.073 (-0.002 to 0.15) | 0.064 (-0.014 to 0.14) | -0.022 (-0.090 to 0.047) | 0.15 (0.062 to 0.23) |
| | P = 0.056 | P = 0.11 | P = 0.54 | P = $7.2 \times 10^{-4}$ |
| VIM of DBP | 0.064 (-0.012 to 0.14) | 0.10 (0.025 to 0.18) | 0.010 (-0.060 to 0.080) | 0.090 (0.0045 to 0.18) |
| | P = 0.098 | P = $9.8 \times 10^{-3}$ | P = 0.77 | P = 0.039 |
| ARV of DBP | 0.083 (0.008 to 0.16) | 0.11 (0.036 to 0.19) | -0.009 (-0.078 to 0.060) | 0.13 (0.045 to 0.21) |
| | P = 0.030 | P = $4.3 \times 10^{-3}$ | P = 0.80 | P = $2.7 \times 10^{-3}$ |
| Diastolic dipping (continuous) | 0.032 (-0.043 to 0.11) | 0.047 (-0.031 to 0.13) | 0.032 (-0.037 to 0.10) | -0.014 (-0.10 to 0.071) |
| | P = 0.40 | P = 0.24 | P = 0.36 | P = 0.74 |
| Binary diastolic dipping (non-dippers versus dippers) | -0.22 (-0.52 to 0.084) | -0.21 (-0.52 to 0.11) | -0.15 (-0.43 to 0.13) | -0.037 (-0.38 to 0.31) |
| | P = 0.16 | P = 0.20 | P = 0.28 | P = 0.83 |

LVMi = left ventricular mass indexed to height$^{2.7}$, LADi = left atrial diameter indexed to height, LVIDD = left ventricular internal diameter during diastole, RWT = relative wall thickness. SBP = systolic blood pressure, SDdn = standard deviation weighted for day and night, VIM = variability independent of the mean, ARV = average real variability, DBP = diastolic blood pressure.

Analysis of multiply imputed data. Adjusted for sex, age at outcome assessment; maternal age at delivery, education, parity, and maternal pre-pregnancy BMI; household social class; smoking at age 17; minutes of moderate to vigorous physical activity at age 15; DXA-determined fat mass, height and height$^2$ at age 17. Regression coefficients for continuous exposures are standardised, i.e. they represent the change in SDs of the outcome (cardiac structure measurement) per one SD higher blood pressure.

to 24-hour mean SBP. The 24-hour mean SBP also showed a positive association with RWT (β = 0.18, 95% CI 0.089 to 0.26, P = $8.1 \times 10^{-5}$), with similar magnitudes of association seen for

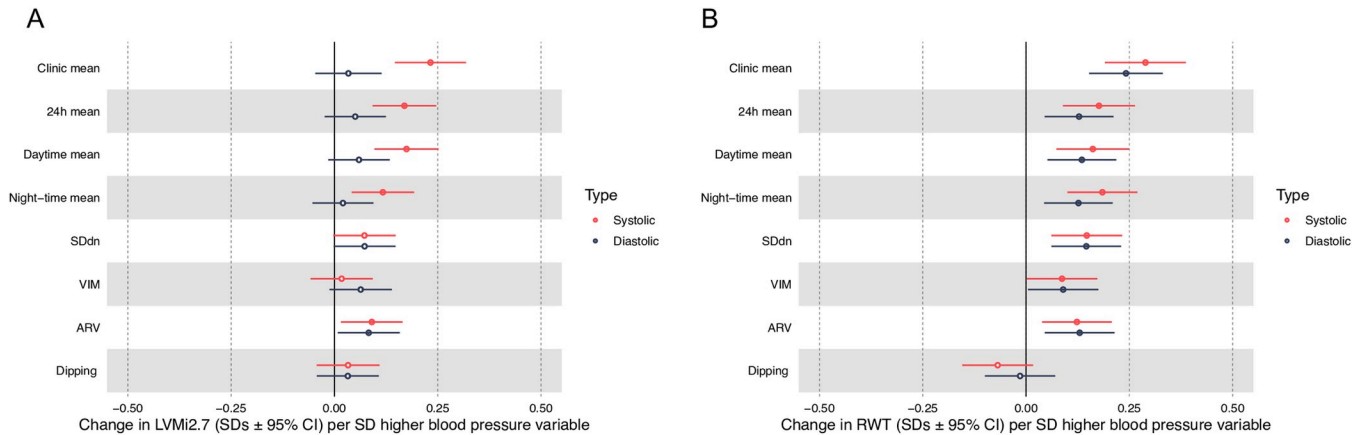

**Fig 2. Forest plot of the mean difference in echocardiographic structures (SDs) per SD higher blood pressure (BP) variable in the confounder model.** A) Left ventricular mass indexed to height $^{2.7}$ (LVMi$^{2.7}$) B) Relative wall thickness (RWT). SDdn = Standard deviation weighted for day and night, VIM = variability independent of the mean, ARV = average real variability. Filled estimates where p<0.05.

daytime and night-time mean SBP (Fig 2B). There was no evidence of associations between 24-hour, day-time or night-time mean SBP and LADi or LVIDD.

There was evidence for associations between all 24-hour DBP measures (mean, day and night) and RWT, with similar magnitudes of associations between the three exposures Fig 2B, but no evidence of associations for the other measures of cardiac structure.

## Associations between 24-hour blood pressure variability and cardiac structures

ARV of SBP was associated with LVMi$^{2.7}$ after adjustment for confounders (Table 2). All three measures of SBP variability (SDdn, ARV, VIM) were positively associated with RWT (SBP SDdn and RWT: β = 0.15, 95% CI 0.061 to 0.23, P = 7.9x10$^{-4}$). There was no consistent evidence of associations between SBP variability and LADi or LVIDD.

DBP variability measures were positively associated with RWT, although evidence of association was weaker for VIM than for SDdn and ARV. ARV and VIM of DBP were also positively associated with LVMi$^{2.7}$ and LADi.

After further adjustment for 24-hour BP (Table 3), associations of SBP and DBP variability with LVMi$^{2.7}$ and RWT attenuated towards the null. Before adjustment for mean DBP the standardised association between ARV of DBP and RWT was 0.13 (95% CI 0.045 to 0.21, P = 2.7x10$^{-3}$). After adjustment for mean DBP the association remained (β = 0.11, 95% CI 0.022 to 0.19, P = 0.014).

## Associations between night-time BP dipping and cardiac structures

The results provided no evidence for associations between either of the dipping variables (percentage difference and categorical) and cardiac structures (S4 Table in S1 File and Tables 2 and 3). This was true for both SBP and DBP.

## Complete case analysis

For all analyses, there were similar magnitudes of estimates between the complete cases and imputed analyses (S2 Table in S1 File and Table 3). However, as there was less power in the complete case analysis, confidence intervals were wider.

**Table 3. Associations of BP variability and dipping with cardiac structure after adjustment for 24-hour mean BP, N = 587.**

| Exposure | Mean difference in cardiac structure measures (SDs) per SD higher BP: β, 95% confidence interval, P value | | | |
|---|---|---|---|---|
| | LVMi | LADi | LVIDD | RWT |
| SDdn SBP | 0.020 (-0.059 to 0.099) | 0.024 (-0.060 to 0.11) | -0.027 (-0.10 to 0.046) | 0.10 (0.011 to 0.19) |
| | P = 0.62 | P = 0.58 | P = 0.47 | P = 0.028 |
| VIM of SBP | 0.020 (-0.054 to 0.094) | 0.031 (-0.048 to 0.11) | -0.022 (-0.091 to 0.047) | 0.090 (0.0054 to 0.17) |
| | P = 0.59 | P = 0.44 | P = 0.53 | P = 0.037 |
| ARV of SBP | 0.044 (-0.034 to 0.12) | 0.076 (-0.0065 to 0.16) | 0.0060 (-0.066 to 0.078) | 0.077 (-0.011 to 0.17) |
| | P = 0.27 | P = 0.071 | P = 0.87 | P = 0.085 |
| Systolic dipping (continuous) | 0.044 (-0.031 to 0.12) | 0.038 (-0.042 to 0.12) | 0.061 (-0.009 to 0.13) | -0.057 (-0.14 to 0.028) |
| | P = 0.35 | P = 0.35 | P = 0.086 | P = 0.19 |
| Binary systolic dipping (non-dippers versus dippers) | -0.14 (-0.31 to 0.041) | -0.072 (-0.26 to 0.12) | -0.043 (-0.21 to 0.12) | -0.039 (-0.24 to 0.16) |
| | P = 0.13 | P = 0.45 | P = 0.61 | P = 0.71 |
| SDdn DBP | 0.066 (-0.01 to 0.14) | 0.081 (0.0013 to 0.16) | -0.012 (-0.082 to 0.058) | 0.13 (0.41 to 0.21) |
| | P = 0.089 | P = 0.046 | P = 0.74 | $P = 3.7 \times 10^{-3}$ |
| VIM of DBP | 0.076 (-0.0007 to 0.15) | 0.093 (0.012 to 0.17) | -0.0002 (-0.071 to 0.070) | 0.12 (0.033 to 0.20) |
| | P = 0.052 | P = 0.024 | P = 0.99 | $P = 6.9 \times 10^{-3}$ |
| ARV of DBP | 0.076 (-0.0007 to 0.15) | 0.14 (0.055 to 0.21) | 0.0021 (-0.068 to 0.072) | 0.11 (0.022 to 0.19) |
| | P = 0.052 | $P = 9.2 \times 10^{-4}$ | P = 0.95 | P = 0.014 |
| Diastolic dipping (continuous) | 0.031 (-0.044 to 0.11) | 0.050 (-0.028 to 0.13) | 0.034 (-0.034 to 0.10) | -0.020 (-0.10 to 0.065) |
| | P = 0.42 | P = 0.21 | P = 0.33 | P = 0.65 |
| Binary diastolic dipping (non-dippers versus dippers) | -0.24 (-0.54 to 0.068) | -0.18 (-0.50 to 0.13) | -0.14 (-0.4 to 0.14) | -0.076 (-0.42 to 0.27) |
| | P = 0.13 | P = 0.25 | P = 0.33 | P = 0.66 |

LVMi = left ventricular mass indexed to height$^{2.7}$, LADi = left atrial diameter indexed to height, LVIDD = left ventricular internal diameter during diastole, RWT = relative wall thickness. SBP = systolic blood pressure, DBP = diastolic blood pressure, SDdn = standard deviation weighted for day and night, VIM = variability independent of the mean, ARV = average real variability.

Analysis of multiply imputed data. Adjusted for sex, age at outcome assessment; maternal age at delivery, education, parity, and maternal pre-pregnancy BMI; household social class; smoking at age 17; minutes of moderate to vigorous physical activity at age 15; DXA-determined fat mass, height and height$^2$ at age 17; mean 24-hour blood pressure (systolic or diastolic, as appropriate for the exposure). Regression coefficients are standardised, i.e. they represent the change in SDs of the outcome (cardiac structure measurement) per one SD higher blood pressure.

## Sensitivity analysis

A likelihood ratio test was performed to compare the association between average blood pressure (in quintiles as both a categorical and continuous variable) and blood pressure variability. Results suggest that this relationship is nonlinear (S3 Table in S1 File). The same was performed for blood pressure/blood pressure variability and cardiac structure outcomes, which suggested that these associations are indeed linear (S3 Table in S1 File). Regression models were then repeated for exposures related to blood pressure variability and dipping, but adjusting for categorical quintiles of blood pressure rather than continuous. Estimates from this analysis (S5 Table in S1 File) were broadly similar to Table 3. Correlations between mean BP and BP variability were low to moderate, with the correlation coefficient (r) ranging from 0.01 to 0.36 (S6 Table in S1 File), suggesting collinearity is not a concern.

## Discussion

In this cross-sectional study of a general population of adolescents, we explored the association between both blood pressure variability and dipping and measures of cardiac structure. The study provided evidence to confirm positive associations between both average 24-hour and

clinic blood pressure measurements with cardiac structures such as RWT, for both systolic and diastolic measures. There was evidence that 24-hour variability measures (SDdn and ARV) were positively associated with RWT, with ARV also showing a positive association with LVMi$^{2.7}$. After adjustment for 24-hour mean BP, evidence remained for an association between ARV of DBP and RWT. No associations were detected between night-time dipping and cardiac structures in this cohort.

Higher mean SBP was associated with higher LVMi$^{2.7}$ and RWT in our study. This observation, together with our previous finding that higher BMI is causally related to higher LV mass [14], suggests that higher values of LVMi$^{2.7}$ and RWT are, on average, related to adverse cardiac remodeling even in this young population, rather than due to high levels of fitness. Previous studies indicate that risk of subclinical organ damage is raised in participants who are pre-hypertensive, reinforcing the importance to consider BP as a risk factor even in those within the "physiologic" BP range [41,42]. Further, these findings support the notion that the influence of BP on cardiac structure may begin early in life [43] and that earlier risk assessment in adolescence may help avoid subclinical organ damage.

Two previous studies, restricted to hypertensive children, did not find an association between 24-hour BP variability and LVMi$^{2.7}$ [16,44]. However, It is possible that these studies did not have sufficient power to detect associations. To our knowledge, this is the first study to explore these associations in a general population cohort of adolescents. Our results indicate that greater BP variability is related to the risk of cardiac remodeling once average BP is accounted for. This was suggested by the observed positive association between measures of DBP variability and RWT. The magnitude of increase in RWT as a result of higher DBP variability could shift the participant's category from being considered to have normal left heart geometry to having geometric remodeling. Furthermore, without intervention during adolescence and early adulthood, these differences might be expected to further widen at older ages, emphasizing the need for prevention. It has been suggested that higher BP variability may lead to organ damage by reduced ability of baroreceptors to modulate blood pressure [45] or due to increased arterial stiffness [46]. However, as we did not assess causality, we cannot rule out that cardiac remodeling may be causing the increase in blood pressure variability. Further studies are warranted to explore mechanisms and direction of causality.

We found no convincing evidence for an association between non-dipping and cardiac structure in young people. There is not strong evidence for an association between night-time dipping and LVMi in hypertensive children [27,47,48]. Most studies finding a positive association between non-dipping and LV mass have been conducted in hypertensive individuals [49]. The majority of the participants in our sample had blood pressures in the normotensive range; other studies which included such participants have also not found evidence of an association [50].

Both DBP and SBP were associated with RWT to a similar extent. However, unlike SBP, DBP did not show associations with LVMi$^{2.7}$. This could reflect a greater importance of systolic pressure (and by implication pulse pressure on LV mass). It may also be at least partially driven by regression dilution bias [51], which is the biasing of the regression slope to towards zero because of the greater levels measurement error for DBP compared with SBP [52].

The current study has several limitations. It is possible that our study may have lacked statistical power to detect some associations between BP variability and dipping and cardiac structure independently of mean BP due to a modest sample size. Despite this limitation, this study provides highly detailed measurements from each participant, and is still comparatively a large cohort size (compared to other studies which utilize ABPM and echocardiography techniques). The lack of longitudinal data is also a limitation; following up the BP variability and cardiac structure at older ages will enable more detailed analyses of how these processes

develop over time. Furthermore, the cohort are of European ancestry and in a localised area of the UK, which may limit its generalisability. The study uses cross-sectional data, which limits our ability to determine the true direction of the association between blood pressure and cardiac structures, and whether this relationship may be causal. The participants included in our analysis are more affluent than the full ALSPAC cohort [19]. However, whilst this does affect the generalisability of the study, it does not necessarily lead to bias in the estimates of associations. ABPMs have been reported to affect sleep quality due to cuff inflation. This may affect night time dipping levels and therefore weaken associations [53]. Additionally, we were not able to assess longer term blood pressure variability, including visit-to-visit variability, which may be another meaningful value in adolescents to predict adult hypertension [43].

Our results show that in adolescents higher clinic and 24-hour BP, as well as an increase in blood pressure variability, are associated with adverse cardiac remodeling. Our study implies that measurement of BP variability might add to the assessment of cardiac remodeling risk in adolescents. It would be valuable to explore whether BP variability and dipping in adolescents track across the life course, and whether these BP measurements in adolescents are predictive of longer-term cardiovascular outcomes.

## Supporting information

**S1 File. Supplementary tables contain additional information including details of missingness, complete case analyses and collinearity analyses.**
(XLSX)

## Acknowledgments

We are extremely grateful to all the families who took part in this study, the midwives for their help in recruiting them, and the whole ALSPAC team, which includes interviewers, computer and laboratory technicians, clerical workers, research scientists, volunteers, managers, receptionists and nurses. We thank Kirsten Leyland for her support with analyses.

## Author Contributions

**Conceptualization:** Daniel Van De Klee, Laura D. Howe.

**Data curation:** Daniel Van De Klee, Nishi Chaturvedi, George Davey Smith, Alun D. Hughes, Laura D. Howe.

**Formal analysis:** Lucy J. Goudswaard, Sean Harrison, Alun D. Hughes, Laura D. Howe.

**Funding acquisition:** Laura D. Howe.

**Investigation:** Lucy J. Goudswaard, Debbie A. Lawlor.

**Methodology:** Lucy J. Goudswaard, Sean Harrison, Debbie A. Lawlor, George Davey Smith, Alun D. Hughes, Laura D. Howe.

**Resources:** Laura D. Howe.

**Supervision:** Laura D. Howe.

**Visualization:** Lucy J. Goudswaard.

**Writing – original draft:** Lucy J. Goudswaard, Laura D. Howe.

**Writing – review & editing:** Lucy J. Goudswaard, Sean Harrison, Nishi Chaturvedi, Debbie A. Lawlor, George Davey Smith, Alun D. Hughes, Laura D. Howe.

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
