## [Decision Letter · Decision Letter 0]

30 Mar 2021

PONE-D-21-06952

Blood pressure variability and night-time dipping assessed by 24-hour ambulatory monitoring: cross-sectional association with cardiac structure in adolescents

PLOS ONE

Dear Dr. Goudswaard,

Thank you for submitting your manuscript to PLOS ONE. After careful consideration, we feel that it has merit but does not fully meet PLOS ONE’s publication criteria as it currently stands. Therefore, we invite you to submit a revised version of the manuscript that addresses the points raised during the review process.

ACADEMIC EDITOR: All issues raised by expert are required.

We look forward to receiving your revised manuscript.

Kind regards,

Vincenzo Lionetti, M.D., PhD

Academic Editor

PLOS ONE

Journal Requirements:

'None'

Additional Editor Comments (if provided):

Reviewers' comments:

Reviewer's Responses to Questions

**Comments to the Author**

1. Is the manuscript technically sound, and do the data support the conclusions?

Reviewer #1: Partly

Reviewer #2: Yes

2. Has the statistical analysis been performed appropriately and rigorously? 

Reviewer #1: Yes

Reviewer #2: Yes

3. Have the authors made all data underlying the findings in their manuscript fully available?

Reviewer #1: Yes

Reviewer #2: No

4. Is the manuscript presented in an intelligible fashion and written in standard English?

Reviewer #1: Yes

Reviewer #2: Yes

5. Review Comments to the Author

Reviewer #1: Goudswaard et al. explored the associations between several blood pressure measurements (clinic, 24-hours, 24-hour variability, night dipping) with echocardiographic indexes of cardiac structure in a population of adolescents. They found that systolic blood pressure, diastolic blood pressure and blood pressure variability are associated with RWT. Systolic blood pressure is also associated with systolic with LVMi2.7. The importance of exploring these associations in a population of adolescent is relevant, being primary prevention a fundamental tool nowadays to prevent healthcare system burdening. However, the paper has some issues that should better be addressed.

Major

As stated in the limitations, the rather small sample size might limit the relevance of the findings. Indeed, the healthy phenotype represented in the cohort might further hinder the appreciation of a concrete effect. The cross-sectional setting is another major limitation, as recognized by the authors. The associations are not strong and, from a clinical point of view, their relevance is mild.

Moreover, the discussion is rather plain and sometimes unspecific (e.g. line 352 “some associations”, lines 367-368 “adverse cardiovascular health” is a quite strong claim but it is not substantiate enough in the text, lines 380-385 are vague and do not draw a conclusion, maybe they should be partially shifted to methods) not emphasizing enough the main findings of the study and not providing neither a definite pathophysiologic hypothesis nor conveying a clear clinical message.

The topic, i.e. primary/primordial prevention, even more in youth, is relevant so the findings deserve to be discussed, but with a more convincing tone. I believe that the authors should meaningful reassess the discussion: first briefly summarizing their main results and then discussing them stressing the importance of earlier prevention/correct risk assessment in adolescence to avoid subclinical organ damage in older age and the importance of not underestimate values within the “physiologic” range (see, for example, PMID: 32936272, PMID: 29908829), maybe contextualizing them with what observed in the current literature (e.g. with some numeric direct confrontation from what observed in other studies in adults).

Minor

•Title: the title is quite generic and not informative of the results of the study.

•Methods: as stated above, the sample size is rather small. I would appreciate a sample size analysis included in the paper.

•Methods: ethnicity has not been adopted as confounder, though it is related with both blood pressure variability and cardiac structure (e.g. PMC7670766, PMID: 19578033, PMID: 31072636).

•Results. Mean population BMI of the population should be reported.

•Figure 2. Though most the associations are found with RWT, Figure 2 only shows LVMi. RWT should be implemented here.

Reviewer #2: The Authors explored the relationship between blood pressure (BP) variability and dipping and left ventricular (LV) remodeling in 587 adolescents (mean age 18 years). The main findings are that an increased BP variability is associated with higher relative wall thickness, and then concentric LV remodeling, even after adjustment for mean BP. The Authors conclude that "Measurement of BP variability might benefit cardiovascular risk assessment in adolescents". These results are quite novel and the results are reasonable. On the other hand, there are some points to consider.

The possible mechanisms relating increased BP variability to LV remodeling should be evaluated in detail.

The relatively small cohort size (particularly considering the usually initial LV remodeling in adolescents) and the lack of longitudinal data are intrinsic limitations of this study, which should be acknowledged and discussed.

To adjust for mean BP, you should exclude multicollinearity between metrics of BP variability and mean BP. I cannot find this evaluation in your Methods section.

I would not speak of cardiovascular risk assessment, but rather of the evaluation of the risk for cardiac remodeling.

6. PLOS authors have the option to publish the peer review history of their article (what does this mean?). If published, this will include your full peer review and any attached files.

Reviewer #1: No

Reviewer #2: No

---

## [Author Response · Author response to Decision Letter 0]

12 May 2021

We thank the reviewers for their comments on our manuscript titled “Blood pressure variability and night-time dipping assessed by 24-hour ambulatory monitoring: cross-sectional association with cardiac structure in adolescents”. We provide a point-by-point response in the document titled "Response_to_reviewers", with line numbers referred to from the tracked changes version of the manuscript attached.

---

## [Decision Letter · Decision Letter 1]

31 May 2021

Blood pressure variability and night-time dipping assessed by 24-hour ambulatory monitoring: cross-sectional association with cardiac structure in adolescents

PONE-D-21-06952R1

Dear Dr. Goudswaard,

We’re pleased to inform you that your manuscript has been judged scientifically suitable for publication and will be formally accepted for publication once it meets all outstanding technical requirements.

Kind regards,

Vincenzo Lionetti, M.D., PhD

Academic Editor

PLOS ONE

Additional Editor Comments (optional):

Reviewers' comments:

Reviewer's Responses to Questions

**Comments to the Author**

1. If the authors have adequately addressed your comments raised in a previous round of review and you feel that this manuscript is now acceptable for publication, you may indicate that here to bypass the “Comments to the Author” section, enter your conflict of interest statement in the “Confidential to Editor” section, and submit your "Accept" recommendation.

Reviewer #1: All comments have been addressed

Reviewer #2: All comments have been addressed

2. Is the manuscript technically sound, and do the data support the conclusions?

Reviewer #1: Yes

Reviewer #2: Yes

3. Has the statistical analysis been performed appropriately and rigorously? 

Reviewer #1: Yes

Reviewer #2: Yes

4. Have the authors made all data underlying the findings in their manuscript fully available?

Reviewer #1: Yes

Reviewer #2: Yes

5. Is the manuscript presented in an intelligible fashion and written in standard English?

Reviewer #1: Yes

Reviewer #2: Yes

6. Review Comments to the Author

Reviewer #1: The authors have adequately addressed all the raised issues, taking a good effort in revising the manuscript. For what concern the title, I appreciate the one proposed by the authors, being more accurate and descriptive, thus I support its change. In general, the paper is now, in my opinion, more precise and suitable for publication.

Reviewer #2: The Authors have modified their manuscript according to my comments. I have no further observations.

7. PLOS authors have the option to publish the peer review history of their article (what does this mean?). If published, this will include your full peer review and any attached files.

Reviewer #1: No

Reviewer #2: No

---

## [Editor Report · Acceptance letter]

8 Jun 2021

PONE-D-21-06952R1 

Blood pressure variability and night-time dipping assessed by 24-hour ambulatory monitoring: cross-sectional association with cardiac structure in adolescents 

Dear Dr. Goudswaard:

I'm pleased to inform you that your manuscript has been deemed suitable for publication in PLOS ONE. Congratulations! Your manuscript is now with our production department. 

Kind regards, 

on behalf of

Prof. Vincenzo Lionetti 

Academic Editor

PLOS ONE